# Leveraging Genomic and Bioinformatic Analysis to Enhance Drug Repositioning for Dermatomyositis

**DOI:** 10.3390/bioengineering10080890

**Published:** 2023-07-27

**Authors:** Lalu Muhammad Irham, Wirawan Adikusuma, Anita Silas La’ah, Rockie Chong, Abdi Wira Septama, Marissa Angelina

**Affiliations:** 1Faculty of Pharmacy, Universitas Ahmad Dahlan, Yogyakarta 55164, Indonesia; 2Research Centre for Pharmaceutical Ingredients and Traditional Medicine, National Research and Innovation Agency (BRIN), South Tangerang 15314, Indonesia; 3Department of Pharmacy, University of Muhammadiyah Mataram, Mataram 83127, Indonesia; 4Research Center for Vaccine and Drugs, National Research and Innovation Agency (BRIN), South Tangerang 15314, Indonesia; 5Taiwan International Graduate Program in Molecular Medicine, National Yang Ming Chiao Tung University and Academia Sinica, Taipei 112304, Taiwan; 6Department of Chemistry and Biochemistry, University of California, Los Angeles, CA 90095, USA

**Keywords:** dermatomyositis, drug discovery, genomic variants, drug repositioning

## Abstract

Dermatomyositis (DM) is an autoimmune disease that is classified as a type of idiopathic inflammatory myopathy, which affects human skin and muscles. The most common clinical symptoms of DM are muscle weakness, rash, and scaly skin. There is currently no cure for DM. Genetic factors are known to play a pivotal role in DM progression, but few have utilized this information geared toward drug discovery for the disease. Here, we exploited genomic variation associated with DM and integrated this with genomic and bioinformatic analyses to discover new drug candidates. We first integrated genome-wide association study (GWAS) and phenome-wide association study (PheWAS) catalogs to identify disease-associated genomic variants. Biological risk genes for DM were prioritized using strict functional annotations, further identifying candidate drug targets based on druggable genes from databases. Overall, we analyzed 1239 variants associated with DM and obtained 43 drugs that overlapped with 13 target genes (*JAK2*, *FCGR3B*, *CD4*, *CD3D*, *LCK*, *CD2*, *CD3E*, *FCGR3A*, *CD3G*, *IFNAR1*, *CD247*, *JAK1*, *IFNAR2*). Six drugs clinically investigated for DM, as well as eight drugs under pre-clinical investigation, are candidate drugs that could be repositioned for DM. Further studies are necessary to validate potential biomarkers for novel DM therapeutics from our findings.

## 1. Introduction

Dermatomyositis (DM) is a rare disease that leads to chronic skin and muscle inflammation, classified as a type of Idiopathic Inflammatory Myopathy [1]. DM is highly prevalent in Asian populations [2,3] and most common in women compared to men between the ages of 40 and 50 years [4]. The etiology of DM involves genetics, immunologic, and environmental factors [1]. For instance, DM has been genetically linked to patients with certain human leukocyte antigen (HLA) types [1]. Some haplotypes associated with high risk include *HLA-A*68* in North American Whites [5], *HLA-DRB1*0301* in African Americans [6], and *HLA-DQA1*0104* and *HLA-DRB1*07* in Han Chinese [7].

Several symptoms of DM include muscle weakness, myalgia, periungual telangiectasias, dystrophic cuticles, and a reddish rash on the heliotrope around the eyes [8]. In particular, a severe symptom of DM is dystrophic calcinosis, which is the deposition of calcium in the soft tissue of DM patients. This is a very painful condition that commonly affects children and adolescents but is rare in adults [9]. Calcinosis develops within 3 years of diagnosis due to delayed diagnosis, insufficient or resistance to treatment, long untreated duration, and severe disease course [9,10]. Calcium channel blockers, especially non-dihydropyridine such as diltiazem, have been beneficial in managing calcinosis. Furthermore, prednisone, azathioprine, and methotrexate have often been used in DM patients [11]. Considering the severity of DM, these treatment approaches have been in use but there is still no cure for DM.

Extensive investigation has also been carried out towards improving DM. However, no proven drugs are currently available to halt the progression of DM. It is important to note that the discovery of new drugs is an extremely costly, high-risk, and time-consuming process [12]. Considering the process of drug development, bringing a new drug to the market is estimated to take around 15 years with more than USD 1 billion [13]. With that in mind, the concept of drug repurposing approaches offers a great opportunity to identify a new drug candidate in a shorter time frame, and with a lower cost in comparison with the complete discovery of a new drug candidate. In light of this, the use of drug repositioning has been known to enable the identification of new indications for existing drugs, and could be a promising strategy for intractable diseases such as DM.

Currently, genomic approaches are beginning to be widely adopted even for rare diseases, due to the availability of several genomic tools to identify genetic markers, resulting in disease prediction and drug discovery. Some genomic tools and databases include genome-wide association study (GWAS) and phenome-wide association study (PheWAS) catalogs. These databases are used to provide multiple risk loci for various diseases including DM. GWAS and PheWAS catalog databases are a rich source of genetic variants associated with diseases such as DM. However, the clinical implementation that involves the translation of valuable biological insight into biological risk genes is limited.

In the present study, we integrated genomic variants involved in DM by using a strict bioinformatics approach. We also applied the functional annotation-driven biological insight based on molecular mechanisms and genetic linkage for DM. Finally, we identified a short list of potential candidate drugs to be repositioned for DM.

## 2. Results

First, we retrieved genomic variants associated with DM from GWAS and PheWAS catalogs. Secondly, we prioritized the DM risk genes based on seven strict functional annotations. Third, we applied network analysis for DM biological risk genes. Finally, a prioritized list of drugs was obtained for DM using drug databases.

### 2.1. Variants Associated with Dermatomyositis from GWAS and PheWAS Catalogs

The current study for DM focused on the application of two widely genomic databases, including GWAS and PheWAS catalogs, to identify functional genomic variants. We found 3 SNPs from the GWAS catalog which were significantly associated with DM (odds ratio (OR) > 1 and *p*-value < 5 × 10^−8^) and 49 associated single nucleotide polymorphisms (SNPs) from the PheWAS catalog (OR > 1 and *p*-value < 0.05). We further expanded the genomic variants based on the neighborhood with LD > 0.8 to filter the same characteristics among variants. Finally, we identified 1239 SNPs and found 78 genes that were encoded by the variants. We further prioritized the biological DM risk genes based on the filters from the scoring system.

### 2.2. Functional Annotation of Dermatomyositis Risk Genes

Seven biological functional annotations were used to prioritize the biological risk genes for DM. One point was awarded for each functional annotation. Scores were assigned to each candidate gene using the following seven criteria: (1) gene variation with missense mutation (*n* = 7); (2) gene variations that have a risk for *cis* expression quantitative trait locus (*cis*-eQTL) (*n* = 18); (3) genes that overlap with the Kyoto Encylopedia of Genes and Genomes (KEGG) (*n* = 5); (4) biological processes (*n* = 5); (5) cellular components (*n* = 2); (6) molecular functions (*n* = 9); and (7) biological risk genes that overlap with Primary Immunodeficiency (PID) (*n* = 2) (Table 1 and Figure 1). Variants were first annotated in order of priority of missense (or nonsense), synonymous, or non-coding mutations. In particular, we mapped genetic variants to corresponding genes with missense/loss-of-function (LoF) mutations, as these non-synonymous changes in a single base substitution can have a significant impact on protein expression. We then used eQTLs, which are regions in the genome that are associated with changes in gene expression, to identify variants that could potentially cause changes in gene expression in the direction of the tissues involved in DM (i.e., whole blood and skin). Furthermore, we utilized PPIs to understand the relationships between diseases and biological protein networks. If the genes involved in these networks are related to DM pathogenesis, inhibiting their protein could be a potential drug repurposing strategy. We also applied knockout mouse phenotypes and KEGG pathways to identify the molecular pathways enriched on the DM-associated gene list and the genes involved. Finally, we incorporated PID diseases, which are innate immune diseases that have been associated with DM, to identify genes that play a causal role in the disease.

Next, we scored each gene based on the number of criteria met (scores from 0 to 7 for each gene) (Figure 1A,B). In order to avoid overlapping between functional annotations, a correlation coefficient analysis was performed. It is more likely that functional annotations will overlap if the value is close to one. Figure 1C depicts the result of the seven functional annotations with values between 0.2 and 0.6, indicating no overlapping between each functional annotation. We found 44 genes with a score of 0, 25 biological DM genes for threshold score ≥ 1, 7 biological DM genes for threshold score ≥ 2, and 3 genes with threshold score ≥ 3. To be considered biological risk genes, we required a score of more than 2, which was defined as “biological dermatomyositis risk genes” (Figure 1D). As shown in Table 1, 10 of the biological DM risk genes are Z-DNA Binding Protein 1 (*ZBP1*), Signal transducer and activator of transcription 2 (*STAT2*), Cluster of Differentiation 247 (*CD247*), Par-3 Family Cell Polarity Regulator Beta (*PARD3B*), Solute Carrier Family 41 Member 1 (*SLC41A1*), Small G Protein Signaling Modulator 2 (*SGSM2*), RNA Binding Motif Single Stranded Interacting Protein 3 (*RBMS3*), Telomerase Reverse Transcriptase (*TERT*), Serine Racemase (*SRR*), and Zinc Finger Protein 544 (*ZNF544*).

### 2.3. Gene Network Expansion through Utilization of the STRING Database

Ten biological DM risk genes were developed by using the STRING database (https://string-db.org/ (accessed on 31 July 2022)). Through using the STRING database, we obtained 60 genes as target genes, which were used for further analysis.

### 2.4. Prioritization of Drugs Repurposed for Dermatomyositis

For this, we mapped 60 target genes into drug databases (DrugBank and DGIdb). We found 43 new drug candidates targeting 13 DM biological risk genes based on the mapping in drug databases (Figure 2). Of these candidates, six are currently undergoing clinical trials according to ClinicalTrial.gov (https://clinicaltrials.gov/ (accessed on 1 August 2022). These drug candidates are linked to six biological DM risk genes: *JAK1*, *JAK2*, *IFNAR1*, *IFNAR2*, *FCGR3B*, and *CD4* (Table 2). In total, we identified nine unique combinations of drugs under clinical investigation for the six target genes (Table 2), corresponding to six unique drugs. The six drug candidates are tofacitinib (NCT03002649), baricitinib (NCT05361109), human immunoglobulin G (NCT02728752), antithymocyte immunoglobulin (NCT00010335), interferon alpha-n1 (NCT00533091), and human interferon beta (NCT05192200). According to PubMed analysis (https://pubmed.ncbi.nlm.nih.gov/ (accessed on 4 August 2022), among the 43 drug candidates, 8 of the identified drugs are currently under pre-clinical testing for DM, including ruxolitinib [14], tofacitinib [15], upadacitinib [16], baricitinib [17], filgotinib [18], human interferon beta [19], interferon alfa-2a [20], and interferon beta-1 [21] (Table 3). These eight drugs were associated with four DM risk genes, including *JAK1*, *JAK2*, *IFNAR1*, and *IFNAR2*.

In conclusion, we found 11 new drug candidates (tofacitinib, baricitinib, human immunoglobulin G, antithymocyte immunoglobulin, interferon alfa-n1, human interferon, ruxolitinib, upadacitinib, filgotinib, interferon alfa-2a, and interferon beta-1) for DM which supported both clinical and pre-clinical data. Furthermore, we observed case reports suggesting that Janus kinase inhibitors [22] may be effective in DM, and the effect may be mediated by preventing the observed upregulation of type 1 interferon [12]. We also found 43 drugs that overlapped with 13 target genes (*JAK2*, *FCGR3B*, *CD4*, *CD3D*, *LCK*, *CD2*, *CD3E*, *FCGR3A*, *CD3G*, *IFNAR1*, *CD247*, *JAK1*, and *IFNAR2*). It is important to highlight that these targets not only can be useful as diagnostic biomarkers and for prognosis, but can also drive drug target identification for DM. Finally, our findings revealed genomic variation as a powerful driver for drug repositioning for DM and can potentially be applied to other complex diseases.

## 3. Discussion

Dermatomyositis (DM) is an autoimmune disease characterized by inflammatory features that affect the skin and muscles. Dysregulation of melanogenesis may contribute to the pathogenesis of DM by affecting immune responses, such as antigen presentation, cytokine production, and T-cell activation, which are modulated by melanin. Additionally, autophagy dysfunction has been linked to various autoimmune disorders, including DM, as it is involved in maintaining cellular homeostasis and regulating protein quality control. In DM, autophagy dysfunction can result in the accumulation of protein aggregates and impaired clearance of apoptotic cells, leading to the release of autoantigens and activation of the immune system [23,24]. Therefore, targeting autophagy may be a potential therapeutic strategy for DM.

In this study, we used the genomic databases (GWAS and PheWAS catalogs) to obtain information on DM susceptibility genes and to further prioritize genes that are at risk for DM based on functional annotations. Herein, the use of genetic research to understand disease biology and its application in the clinic represents a useful approach for DM. Moreover, we were able to prioritize and obtain 10 biological DM risk genes based on GWAS and PheWAS database analyses, using strict functional annotation criteria, and through a gene network expansion to obtain candidate DM drug targets.

In particular, we used seven defined biological criteria to prioritize functional genomic variants and to identify the biological DM risk genes. In this study, 43 overlapping drugs were identified with 13 target genes (*JAK2*, *FCGR3B*, *CD4*, *CD3D*, *LCK*, *CD2*, *CD3E*, *FCGR3A*, *CD3G*, *CD247*, *JAK1*, and *IFNAR2*). Based on clinical and preclinical studies, we showed that 11 of these new drug candidates were identified as promising drugs for treating DM.

Notably, we showcased the DM target genes, namely, *JAK1*, *JAK2*, *IFNAR1*, *IFNAR2 CD4*, and *FCGR3B*, corresponding to nine drug-target combinations that could potentially be repositioned for DM. Remarkably, the identified genes are promising targets for the treatment of DM, as they achieved the highest systemic scores on functional annotations from this study. Importantly, we also found several drug candidates currently under either clinical and/or pre-clinical investigations; five of these drugs target the *JAK1* and *JAK2* genes, supporting the clinical and pre-clinical data for DM (*JAK* inhibitors include ruxolitinib, tofacitinib, upadacitinib, baricitinib, and filgotinib). A previous finding has suggested that *JAK* inhibitors reduce skin signs and symptoms and increase muscle strength [25]. Inclusively, it has also been shown that *JAK1* and *JAK2* are related to DM susceptibility [26].

CD4 T lymphocytes play an important role in the pathogenesis of DM by triggering antibodies that repair damaged vascular components [27]. We found that drugs that overlap with CD4 include human interferon beta, interferon alpha-2a, and interferon beta-1a. Meanwhile, *IFNAR1* and *IFNAR2* are IFN-α receptor subunits that affect DM. As *IFNAR* activates IFN-1, it causes muscle and endothelial cell damage, resulting in DM disease [28]. In addition, interferon beta is a drug that overlaps with *IFNAR1* while interferon alpha-1 is a drug that overlaps with *INFAR2*. Thus, we also identified drugs overlapping with *FCGR3B* as Human Immunoglobulin G. According to the mechanism of drug action, this can block the Fc receptors by binding to the inhibitory receptors (FcgR2b) of Fc by activating FcgR1 and FcgR3 receptors, thereby suppressing the antibodies [29]. Of note, the expression of identified target genes that are immune-related systemic genes could be specifically transcribed during inflammation or are specific for different skin cells, and could provide further insights into their potential roles in dermatomyositis [30,31].

It is important to consider both the limitations and the advantages of our approach for drug repositioning for DM. For example, we obtained markedly less genomic variants that encoded our genes of interest. Furthermore, not all the biological risk genes are druggable, thus reducing the number of candidate drugs. We believe that the benefits of this approach outweigh the limitations outlined above because this approach enables the identification of the biological risk genes that could be extended to many other multi-factorial genetic disorders beyond DM. These bioinformatic approaches link the data to the drug database, further narrowing down the candidate drugs for many polygenic diseases, leading to cost and time savings in the drug discovery process.

## 4. Materials and Methods

### 4.1. Workflow for Integrative Analysis of Genomic Variants and Gene Network

A detailed workflow of this study is shown in Figure 3. In this study, we prioritize data on DM-associated SNPs. These variants were obtained from the genomic database, namely, GWAS and PheWAS catalog databases. In the GWAS database, the criteria used for SNPs associated with DM were prioritized based on the *p*-value (<10^−8^) and odds ratio (OR) > 1. In the PheWAS database, the criteria used for SNPs associated with DM were determined based on the *p*-value <0.01 and OR > 1. HaploReg database version 4.1 was used to explore the genomic variants. We expanded the range of adjacent SNPs according to the criterion of r^2^ > 0.8 to obtain more SNPs and genes associated with DM. We realized that the more SNPs that we identified, the more candidate genes were found.

To prioritize the risk gene candidates for DM, we used a scoring system with seven functional annotation criteria. Following the scoring system, a total score equal and greater than two (score ≥ 2) was identified as a DM biological risk gene. In this study, the scoring system was modified based on Okada et al. [32] and applied by Irham et al. for several diseases [33,34,35,36,37,38]. Next, we used the STRING database to obtain additional DM-targetable genes and the expanded biological DM risk gene. Furthermore, we mapped the biological DM risk genes according to the DrugBank database to identify potential drug targets. ClinicalTrials.gov (accessed on 30 October 2022) and PubMed were used to validate the drugs undergoing clinical trials and pre-clinical studies (in vitro and in vivo, respectively).

### 4.2. Candidate Risk Genes Associated with Dermatomyositis

The SNPs associated with DM were obtained from GWAS and PheWAS catalogs. We ensured that all SNPs were unique without duplicated SNPs. SNPs that fulfill the criterion were utilized for further analysis. In this study, we used HaploReg version 4.1 with r^2^ > 0.8 criteria to obtain SNPs encoding for DM-related genes. The SNP-encoded genes were prioritized as the genes associated with DM. HaploReg version 4.1 was used to determine the encoded variant genes and further showed the functional role in the pathogenesis of disease through the affected protein [39].

### 4.3. Biological Risk Genes for Dermatomyositis

This study used seven strict functional annotation criteria with a scoring system to prioritize biological DM risk genes. Based on the criteria, each functional annotation is assigned a score of one. A total score of two or more (score ≥ 2) is required to be classified as a biological risk gene. HaploReg version 4.1 (https://pubs.broadinstitute.org/mammals/haploreg/haploreg.php (accessed on 28 August 2022) was used to determine the criterion for missense mutation; we considered seven functional annotation criteria, namely: (1) we required a missense mutation that was used for functional annotations due to amino acid changes that led to protein function change; (2) *cis*-eQTL was used to determine whether the genetic variants affected protein expression, resulting in gene expression changes towards the involved tissue; (3) KEGG was used to determine the involvement of molecular pathways based on KEGG data with a significance false discovery rate (FDR) *q* < 0.05); (4) the biological process was used to determine the genes involved in the biological protein networks and to determine the prioritized inhibitory protein in the biological processes—we considered FDR *q* < 0.05 as significant; (5) cellular components; (6) molecular functions; and (7) PID gene. PID is an innate immune disease that is shown to be associated with DM pathogenesis. The correlation coefficient analysis was performed to determine whether seven functional annotations have possible linear relationships.

### 4.4. Gene Network Expansion by Using STRING Database

The STRING database was used to integrate publicly accessed sources of information by direct (physical) and indirect (functional) protein–protein interactions. To obtain more potential drug targets, the STRING database (https://string-db.org/ (accessed on 31 July 2022) was utilized to expand the biological DM risk genes.

### 4.5. Gene and Drug Overlapping Analysis from Drug Databases

To obtain new drug targets for DM, overlapping analyses between gene target candidates and drug candidates were conducted using DrugBank (https://go.drugbank.com/ (accessed on 2 August 2022) and the drug–gene interaction database (DGIdb) (https://www.dgidb.org/ (accessed on 3 August 2022)). The requirements for candidate drug targets were: demonstrated pharmacological activity, guaranteed effectiveness, approved annotations, and to have had clinical trials. We used ClinicalTrial.gov (https://clinicaltrials.gov/ (accessed on 1 August 2022) and PubMed (https://pubmed.ncbi.nlm.nih.gov/ (accessed on 4 August 2022) to verify whether each identified new drug is under clinical trials for DM or other diseases.

## 5. Conclusions

The integration of genomic variants and gene network analysis revealed candidate drug targets for dermatomyositis (DM). We analyzed 1239 variants associated with dermatomyositis and obtained 43 drugs that overlapped with 13 target genes (*JAK2*, *FCGR3B*, *CD4*, *CD3D*, *LCK*, *CD2*, *CD3E*, *FCGR3A*, *CD3G*, *IFNAR1*, *CD247*, *JAK1*, *IFNAR2*). Interestingly, six drugs that overlapped with six target genes were clinically investigated for DM and are candidate drugs that could be repositioned for DM. In addition, we found eight drugs currently under a pre-clinical trial that overlapped with the six target genes from our analysis. Overall, this study unveiled novel biological insights to drive drug discovery for DM by integrating genomic variants and gene network analysis.

## Figures and Tables

**Figure 1 bioengineering-10-00890-f001:**
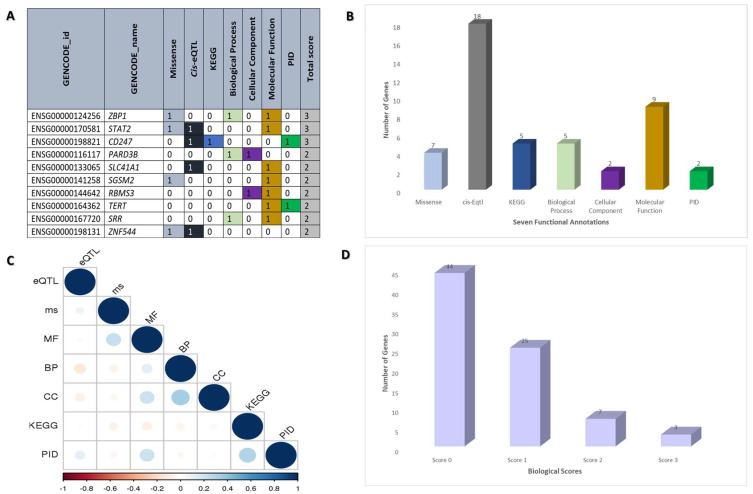
(**A**) Functional bioinformatic annotations were integrated with genomic information to prioritize dermatomyositis (DM) biological risk genes. (**B**) The number of genes for each of the seven biological criteria used for DM risk gene prioritization. (**C**) Correlogram indicating the pairwise Phi correlation coefficient across the seven criteria for DM risk gene prioritization. The blue color denotes a positive correlation while the red color denotes a negative correlation. (**D**) Distribution of scores based on the DM risk gene annotations from the annotation scoring system (0 to 3, with 3 being the highest score), with the number of genes for each score bin indicated.

**Figure 2 bioengineering-10-00890-f002:**
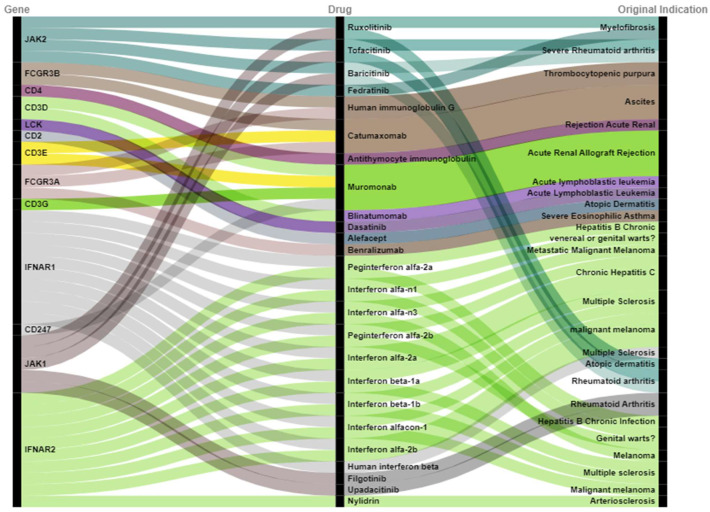
Alluvial diagram showing the 43 drugs overlapped with 13 target genes for DM.

**Figure 3 bioengineering-10-00890-f003:**
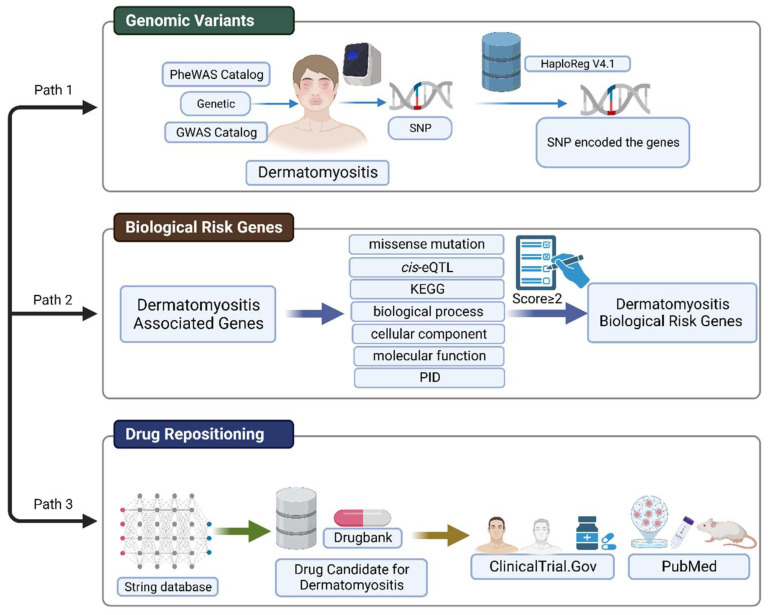
Schematic model illustrating how the genomic variants can be leveraged for drug repositioning in DM. This figure was created with BioRender.com under agreement number “*FQ24UFDG8T*”.

**Table 1 bioengineering-10-00890-t001:** Seven functional annotations were applied to prioritize the biological risk genes for dermatomyositis (DM).

GENCODE ID	Gene Name (GENCODE)	Missense	*Cis*-eQTL	KEGG	Biological Process	Cellular Component	Molecular Function	PID	Total Score
ENSG00000124256	*ZBP1*	1	0	0	1	0	1	0	3
ENSG00000170581	*STAT2*	1	1	0	0	0	1	0	3
ENSG00000198821	*CD247*	0	1	1	0	0	0	1	3
ENSG00000116117	*PARD3B*	0	0	0	1	1	0	0	2
ENSG00000133065	*SLC41A1*	0	1	0	0	0	1	0	2
ENSG00000141258	*SGSM2*	1	0	0	0	0	1	0	2
ENSG00000144642	*RBMS3*	0	0	0	0	1	1	0	2
ENSG00000164362	*TERT*	0	0	0	0	0	1	1	2
ENSG00000167720	*SRR*	0	0	0	1	0	1	0	2
ENSG00000198131	*ZNF544*	1	1	0	0	0	0	0	2
ENSG00000069275	*NUCKS1*	0	1	0	0	0	0	0	1
ENSG00000069667	*RORA*	0	0	1	0	0	0	0	1
ENSG00000103653	*CSK*	0	1	0	0	0	0	0	1
ENSG00000110944	*IL23A*	0	0	1	0	0	0	0	1
ENSG00000112294	*ALDH5A1*	0	1	0	0	0	0	0	1
ENSG00000117280	*RAB7L1*	0	1	0	0	0	0	0	1
ENSG00000128815	*WDFY4*	0	1	0	0	0	0	0	1
ENSG00000128915	*NARG2*	0	1	0	0	0	0	0	1
ENSG00000135469	*COQ10A*	1	0	0	0	0	0	0	1
ENSG00000135823	*STX6*	0	0	0	1	0	0	0	1
ENSG00000135903	*PAX3*	0	0	0	0	0	1	0	1
ENSG00000137261	*KIAA0319*	0	1	0	0	0	0	0	1
ENSG00000139540	*SLC39A5*	0	0	0	0	0	1	0	1
ENSG00000139645	*ANKRD52*	1	0	0	0	0	0	0	1
ENSG00000144785	*RP11-977G19*	0	1	0	0	0	0	0	1
ENSG00000152595	*MEPE*	0	0	0	1	0	0	0	1
ENSG00000160185	*UBASH3A*	0	1	0	0	0	0	0	1
ENSG00000183354	*KIAA2026*	0	1	0	0	0	0	0	1
ENSG00000204287	*HLA-DRA*	0	0	1	0	0	0	0	1
ENSG00000231389	*HLA-DPA1*	0	0	1	0	0	0	0	1
ENSG00000237241	*RP11563N6.4*	0	1	0	0	0	0	0	1
ENSG00000238809	*snoU13*	0	1	0	0	0	0	0	1
ENSG00000245534	*RP11-219B17*	0	1	0	0	0	0	0	1
ENSG00000259462	*RP11-752G15*	0	1	0	0	0	0	0	1
ENSG00000261801	*RP11-941F15*	1	0	0	0	0	0	0	1

**Table 2 bioengineering-10-00890-t002:** Drugs under clinical investigation for DM, with identified drugs and their corresponding target genes for DM.

Gene	Drug	Original Indication	NCT Number
*JAK2*	Tofacitinib	Severe Rheumatoid arthritis	NCT03002649
*JAK2*	Baricitinib	Severe Rheumatoid arthritis	NCT05361109
*FCGR3B*	Human immunoglobulin G	Thrombocytopenic purpura	NCT02728752
*CD4*	Antithymocyte immunoglobulin	Rejection Acute Renal	NCT00010335
*IFNAR1*	Interferon alfa-n1	Genital warts	NCT00533091
*IFNAR1*	Human interferon beta	Multiple Sclerosis	NCT05192200
*JAK1*	Tofacitinib	Rheumatoid arthritis	NCT03002649
*JAK1*	Baricitinib	Rheumatoid arthritis	NCT05361109
*IFNAR2*	Interferon alfa-n1	Genital warts	NCT00533091

**NCT Number**: National Clinical Trial identifier number (ClinicalTrials.gov; accessed on 1 August 2022).

**Table 3 bioengineering-10-00890-t003:** Drugs under pre-clinical investigation and their correspondence with target genes for DM.

Target Gene	Drug	PMID
*JAK1*, *JAK2*	Ruxolitinib	26448614
	Tofacitinib	33258553
	Upadacitinib	35081305
	Baricitinib	35318646
	Filgotinib	32222877
*IFNAR1*, *IFNAR2*	Human interferon beta	27564228
	Interferon alfa-2a	24638953
	Interferon beta-1a	18936398

**PMID**: PubMed identifier.

## Data Availability

Not applicable.

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
