# Peer review of "Leveraging Genomic and Bioinformatic Analysis to Enhance Drug Repositioning for Dermatomyositis"

_bioengineering, 2023, doi:10.3390/bioengineering10080890_

Round 1

Reviewer 1 Report

Authors described a novel approach for drug repositioning for dermatomyositis.

Major comments:

Regarding “Functional annotation of risk genes” - What was the rationale of criteria selection? E.g. Why do the Authors focus on missense but not in the LoF variants category? Why do they not use any other criteria of pathogenicity, like MAF in reference databases, prediction scores etc?

It is completely unclear how the “STRING database (https://string-db.org/) was utilized to expand the biological dermatomyositis risk genes” and how Authors can justify this expansion?

Minor comments:

The number of genes with missense mutations (n=5) is different from the number of genes with missense mutations in Table 1.

“Interestingly, we found 44 genes with a score of 0 and 25 genes with a score of 1. Among these genes, 7 had a score of 2, and 3 had a total score of 3.” <- I do not understand the second sentence.  Among which genes?

“We also dictated 10 genes with a score of more than 2 which were defined as

“biological dermatomyositis risk genes” (Figure 1D). “ 

– I can only see 3 genes, not 10 with a score more than 2. Should it be “score of more than 1”?

“Among the 43 new drug candidates, nine of which are currently undergoing clinical trials for dermatomyositis according to ClinicalTrial.gov (https://clinicaltrials.gov/), the candidate drugs are Tofacitinib(NCT03002649), Baricitinib (NCT05361109), Human immunoglobulin G (NCT02728752), Antithymocyte immunoglobulin (NCT00010335), Interferon alpha-n1 (NCT00533091), and Human interferon beta (NCT05192200). Six drug candidates are linked to six dermatomyositis biological risk genes, including JAK1, JAK2, IFNAR1, IFNAR2, FCGR3B, and CD4 (Figure 3).  “ 

This is also confusing. First Authors mention about 9 drugs under clinical trials and than they list 6 genes related to 6 genes - how these were selected ?

I think that data from Figure 3 and 4 would be easier to follow if presented in the form of Table. 

Author Response

Dear Editors and Reviewers,

Please find our attached revised manuscript, entitled “Leveraging genomic and bioinformatic analysis to enhance drug repositioning for dermatomyositis,” which we are submitting for consideration for publication as Original Research article in Bioengineering-2288341. We consider to submit this paper in the special issue of "Present and Future Therapies of Skin Diseases". This special issue belongs to the section "Biomedical Engineering and Biomaterials". We are thankful for your kind encouragement regarding our manuscript. Herewith, we are sending this revised manuscript in accordance with the comments given by the reviewers. The revised parts of the manuscript are highlighted in yellow. Finally, we would like to thank you once again for giving us the opportunity to improve our manuscript. We very much hope that these revisions are adequate. We appreciate your review and assistance, and look forward to hearing from you. 

Sincerely yours,

Professor Zainul Amiruddin Zakaria

Borneo Research on Algesia,

Inflammation and Neurodegeneration (BRAIN) Group

Department of Biomedical Sciences

Faculty of Medicines and Health Sciences

University Malaysia Sabah

Jalan UMS, Kota Kinabalu 88400, Sabah, Malaysia

Reviewer 2 Report

I think this is a good manuscript that describes an original approach to dermatomyositis genetic mechanisms.

It is a well-presented and well-conducted study.

I do have some comments and suggestions:

1. What s the role of the identified genes in inflammation and skin function? Authors can use these papers PMID: 23515576, 25545474 to identify immune-related systemic genes that are specifically transcribed during the inflammation or are specific for different skin cells. I believe this could be very helpful as dermatomyositis combines skin and systemic muscular pathology.

2. I also suggest authors think or describe the transcribed genes in the context of some specific intracellular functions like autophagy or melanogenesis like in the publications PMID: 18514490, 21879234. Melanogenesis has an impact on inflammation and changes in pigmentation are relevant for dermatomyositis. Autophagy is a common process in complex diseases. These parts would improve the translational impact of the manuscript.

Author Response

Dear Editors and Reviewers

Please find our attached revised manuscript, entitled “Leveraging genomic and bioinformatic analysis to enhance drug repositioning for dermatomyositis,” which we are submitting for consideration for publication as Original Research article in Bioengineering-2288341. We consider to submit this paper in the special issue of "Present and Future Therapies of Skin Diseases". This special issue belongs to the section "Biomedical Engineering and Biomaterials". We are thankful for your kind encouragement regarding our manuscript. Herewith, we are sending this revised manuscript in accordance with the comments given by the reviewers. The revised parts of the manuscript are highlighted in yellow. Finally, we would like to thank you once again for giving us the opportunity to improve our manuscript. We very much hope that these revisions are adequate. We appreciate your review and assistance, and look forward to hearing from you. 

Sincerely yours,

Professor Zainul Amiruddin Zakaria

Borneo Research on Algesia,

Inflammation and Neurodegeneration (BRAIN) Group

Department of Biomedical Sciences

Faculty of Medicines and Health Sciences

University Malaysia Sabah

Jalan UMS, Kota Kinabalu 88400, Sabah, Malaysia
